# Plasma Nitrate and Nitrite Kinetics after Single Intake of Beetroot Juice in Adult Patients on Chronic Hemodialysis and in Healthy Volunteers: A Randomized, Single-Blind, Placebo-Controlled, Crossover Study

**DOI:** 10.3390/nu14122480

**Published:** 2022-06-15

**Authors:** Agustina Heredia-Martinez, Guillermo Rosa-Diez, Jorge R. Ferraris, Anna-Karin Sohlenius-Sternbeck, Carina Nihlen, Annika Olsson, Jon O. Lundberg, Eddie Weitzberg, Mattias Carlström, Rafael T. Krmar

**Affiliations:** 1Department of Nephrology, Hospital Italiano de Buenos Aires, Buenos Aires C1199 CABA, Argentina; agustina.heredia@hospitalitaliano.org.ar (A.H.-M.); guillermo.rosadiez@hospitalitaliano.org.ar (G.R.-D.); jorge.ferraris@hospitalitaliano.org.ar (J.R.F.); 2Department of Chemical and Pharmaceutical Safety, Research Institutes of Sweden (RISE), SE-151 36 Södertälje, Sweden; anna-karin.sternbeck@ri.se (A.-K.S.-S.); 3Department of Physiology and Pharmacology, Karolinska Institutet, Biomedicum 5B, SE-171 77 Stockholm, Sweden; carina.nihlen@ki.se (C.N.); annika.olsson@ki.se (A.O.); jon.lundberg@ki.se (J.O.L.); eddie.weitzberg@ki.se (E.W.)

**Keywords:** dietary nitrate, nitrite, kinetics, beetroot juice, hemodialysis

## Abstract

Nitric oxide (NO) contributes to maintaining normal cardiovascular and renal function. This bioactive signalling molecule is generally formed enzymatically by NO synthase in the vascular endothelium. NO bioactivity can also be attributed to dietary intake of inorganic nitrate, which is abundant in our diet, especially in green leafy vegetables and beets. Ingested nitrate is reduced to nitrite by oral commensal bacteria and further to NO systemically. Previous studies have shown that dialysis, by means of removing nitrate and nitrite from the body, can reduce NO bioactivity. Hence, dietary intervention approaches aimed to boost the nitrate–nitrite–NO pathway may be of benefit in dialysis patients. The purpose of this study was to examine the kinetics of plasma nitrate and nitrite after a single intake of nitrate-rich concentrated beetroot juice (BJ) in adult hemodialysis (HD) patients and in age-matched healthy volunteers (HV). Eight HD patients and seven HV participated in this single center, randomized, single-blind, placebo-controlled, crossover study. Each participant received a sequential single administration of active BJ (70 mL, 400 mg nitrate) and placebo BJ (70 mL, 0 mg nitrate) in a random order separated by a washout period of seven days. For the kinetic analysis, blood samples were collected at different time-points before and up to 44 h after BJ intake. Compared with placebo, active BJ significantly increased plasma nitrate and nitrite levels both in HD patients and HV. The area under the curve and the maximal concentration of plasma nitrate, but not of nitrite, were significantly higher in HD patients as compared with HV. In both groups, active BJ ingestion did not affect blood pressure or plasma potassium levels. Both BJs were well tolerated in all participants with no adverse events reported. Our data provide useful information in planning dietary nitrate supplementation efficacy studies in patients with reduced NO bioactivity.

## 1. Introduction

Cardiovascular disease (CVD) is the leading cause of death in adult patients with maintenance hemodialysis (HD) [1]. Previous studies have also highlighted the substantial risk of CVD mortality in children and young adults on chronic renal replacement treatment [2].

Endogenous nitric oxide (NO) has long been recognized for its vasodilator function, where it plays a key role in the regulation of blood flow, blood pressure (BP), and renal function [3,4]. Indeed, reduced NO production/signaling is largely considered to contribute to the pathogenesis of CVD [3,4]. In addition to the classical L-arginine-dependent NO synthase (NOS) pathway, NO bioavailability can be increased via a serial reduction in the inorganic anions nitrate and nitrite [5,6]. This nitrate–nitrite–NO pathway can be fueled by dietary nitrate intake [7] after which absorbed nitrate is concentrated in the salivary glands, excreted in saliva and reduced to nitrite by oral commensal bacteria. Swallowed nitrite-rich saliva is then further reduced to NO and other nitrogen oxides by several systemic pathways [5]. The nitrate–nitrite–NO pathway has been proposed to be of particular importance during certain disease conditions associated with reduced NOS enzyme activity such as aging and cardiovascular disease as well as ischemia–reperfusion injury [3,7,8]. In previous studies, we have observed that peritoneal dialysis can impair NO homeostasis in children, partially by lowering circulating NO substrates, including L-arginine, nitrate, and nitrite, and downstream signaling of NO [9]. In a recently published clinical study [10], we also found that post-dialysis plasma nitrate, nitrite, and cGMP levels as well as amino acids coupled to NOS activity (i.e., arginine and citrulline) were significantly reduced in adult maintenance hemodialysis (HD) patients.

During the last decade, there has been an increased interest in investigating the health aspects of dietary nitrate supplementation (aimed to increase NO bioactivity) with a particular focus on cardiovascular regulation and exercise performance [3,4,11,12]. With this in mind, we previously hypothesized that dietary nitrate supplementation may be of benefit in boosting the activity of the nitrate–nitrite–NO pathway, thus restoring NO bioactivity in adult HD patients [10]. However, the effects of a high-nitrate supplement on the kinetics of plasma nitrate and nitrite concentrations in HD patients have not been described in detail. Such information would be useful in planning dietary nitrate supplementation efficacy studies in this specific patient population. Consequently, the primary objective of this study was to examine the kinetics of plasma nitrate and nitrite, both in adult patients on chronic HD and in healthy volunteers (HV), after acute ingestion of nitrate-rich concentrated beetroot juice (BJ).

## 2. Material and Methods

### 2.1. Participants and Study Design

Approval for this study was obtained from the Hospital Italiano de Buenos Aires Ethics Committee for human investigations (Protocol number 4043, 4 April 2019). All participants received oral and written information and signed an informed consent form. This study was conducted at the Department of Nephrology of the Hospital Italiano de Buenos Aires, Argentina from 4 November 2019 to 17 May 2021, in accordance with The Principle of Ethics of the World Medical Association that originated in the Declaration of Helsinki for experiments involving humans.

In patients with end-stage renal disease on HD, the inclusion criteria were age 18 years or older and ≥3 months on renal replacement therapy at time of enrollment. Exclusion criteria were diagnosis of type 1 or 2 diabetes and positive serology for HIV-1 or HIV-2 antibody, Hepatitis B surface antigen or Hepatitis C antibody. In HV, the inclusion criteria were age 18 years or older and body mass index between 20 and 29 kg/m^2^, whereas the exclusion criterion was suspicion of ongoing infection on the day of the study assessment.

Our study population consisted of 8 adults with end-stage renal disease (4 males), with a median age of 31 years (range 23–45) that were on maintenance post-dilution hemodiafiltration. All these patients were examined on the morning of their first dialysis session of the week, which represents the larger interval between consecutive dialysis sessions. The control group consisted of 7 HV (3 males) with a median age of 36 years (range 30–40). These apparently healthy controls were staff members at the Department of Nephrology. They were asked to complete a medical history questionnaire, which also included questions on whether routine serum blood chemistry profiles were performed in the year prior to enrollment into the current study. All participants communicated that they had absence of disease, were not taking any medication, and if the information was available, had normal liver and renal function test results (*data not shown*).

All participants completed all aspects of this study and were instructed to abstain from antibacterial mouthwash, which is known from both preclinical and clinical studies to disrupt the ability of reducing nitrate to nitrite [13,14,15,16]. In an attempt to minimize the variability in the measurement of endogenous plasma nitrate and nitrite due to different dietary intake of these anions [17], the participants were asked to adhere to a diet poor in nitrate content, i.e., avoid vegetables and fruits [18] 24 h before and 44 h after each trial.

The design used was a randomized (order of the types of juice, i.e., nitrate-rich and nitrate-depleted beetroot juice (BJ), participant-blind, placebo-controlled, crossover study. Due to the study design, all participants served as their own controls. An adverse event (AE) was defined as any untoward medical occurrence in a participant administered nitrate-rich or placebo BJ and which did not necessarily have a causal relationship with the ingestion of the specific BJ. An AE could therefore be any unfavorable and unintended sign (including an abnormal laboratory finding), symptom, or disease temporally associated with the use of nitrate-rich or placebo BJ, whether or not it was considered related to them.

### 2.2. Composition of Nitrate-Enriched and Placebo Beetroot Juice

All participants were given a single portion of 70 mL nitrate-rich or nitrate-depleted (placebo) BJ (James White Drinks Ltd., Ipswich, UK), in a crossover manner, with a washout period of at least seven days between each trial. The nitrate-rich BJ contained 400 mg nitrate (92.4 ± 11.9 mmol/L; 70 mL), and the placebo BJ contained <0.01 mmol/L nitrate; 70 mL); otherwise, they were identical in taste and appearance [19]. This dose of nitrate, using similar BJ, has previously been shown to have BP lowering effects [20,21].

### 2.3. Blood, Dialysate, and Saliva Collection

Venous blood samples (approximately 2 mL) for measurement of nitrate and nitrite levels were withdrawn before the start and at the end of the dialysis session. The latter represented the baseline sample (0 h), i.e., before the ingestion of nitrate-rich/placebo BJ, which took place at the end of the dialysis session. Thereafter, blood samples were withdrawn 0.25, 0.75, 1.5, 4, 8, and 44 h post-ingestion. In HV, venous blood samples (approximately 2 mL) for concentration measurement of nitrate and nitrite were withdrawn before the ingestion of nitrate-rich/placebo BJ (0 h) and at 0.25, 0.75, 1.5, 4, 8, and 48 h post-ingestion.

All samples were collected in EDTA tubes (5 mM) and processed to plasma (centrifugation at 4500× *g*, 5 min, 4 °C) and subsequently frozen at −70 °C until analysis.

At the end of the HD session, an aliquot (5 mL) derived from a small representative sample of the total spent dialysate (including the ultrafiltrate and the substitution volume during the dialysis session) was obtained to measure nitrate and nitrite concentrations (D_1_). To secure a representative sample of the total spent dialysate, we used the partial spent dialysate collection method as previously described by Ing et al. [22]. In addition, a sample of fresh dialysate was taken before its entry to the dialyzer before the commencement of the dialysis session and analyzed for nitrate and nitrite (D_0_).

Saliva samples were collected in parallel, i.e., before the start of dialysis and at the end of the dialysis session, i.e., before the ingestion of nitrate-rich/placebo BJ as well as 4 h post-ingestion. The saliva was directly collected into 1.5 mL Eppendorf tubes, frozen immediately and stored at −70 °C until analysis. In HV, saliva was collected pre- and 4 h post-ingestion of nitrate-enriched/placebo BJ.

### 2.4. Hemodialysis Procedure

All participants with end-stage renal disease were dialyzed thrice-weekly with a Fresenius 5008 dialysis machine (Fresenius Medical Care, Bad Homburg, Germany) using Helixone^®^-based dialysis membrane (Fresenius S.E., Bad-Homburg, Germany). Post-dilution hemodiafiltration was performed with ultrapure bicarbonate dialysate. The prescribed extracorporeal blood flow rate was ≈300 mL/min, whereas the prescribed dialysate flow rate was set at 500 mL/min. The delivered dialysis dose was determined by the calculation of Kt/V_urea_ [23]. This index of the adequacy of the dialysis treatment indicates the blood volume that has been completely cleared of urea during a specified dialysis. A Kt/V_urea_ ≥ 1.2 is the recommended minimum dose per session [23]. In the present study, the measurement of delivered Kt/V_urea_ was determined automatically at the end of the dialysis by the Online Clearance Monitoring Method [24].

The amount of excreted nitrate and nitrite during the dialysis session was calculated by multiplying the delta concentration of these anions, i.e., D_1_–D_0_, and the volume of the total effluent volume at the end of the HD. The total effluent volume includes the total spent dialysate, ultrafiltrate, and the substitution volume during the treatment time.

### 2.5. Blood Pressure Measurements

All BP measurements were performed by using an automatic oscillometric BP device (module incorporated in the 5008-dialysis machine in HD patients and OMRON HEM 7322T-E BP monitor in HV, respectively) with the participant seated; at each time point, three BP measurements were obtained, and the mean of the 2nd and 3rd reading was used. In HD patients, BP was measured before the start and at the end of dialysis session, i.e., before nitrate-rich/placebo BJ ingestion and 4 h later. In HV, BP was measured before nitrate-rich/placebo BJ ingestion and 4 h later.

### 2.6. Nitrate and Nitrite in Plasma, Saliva, and Dialysate

The plasma samples were extracted using HPLC grade methanol (CROMASOLV, Sigma-Aldrich, Burlington, MA, USA), and the saliva samples were centrifuged at 10,000× *g* for 10 min at 4 °C before dilution with carrier solution containing 10% HPLC grade methanol. Dialysate samples were analyzed undiluted. Subsequently, nitrite and nitrate levels were measured HPLC (ENO-20 Eicom, Kyoto, Japan), which uses reverse phase chromatography to separate nitrite from nitrate. Thereafter, nitrate was reduced to nitrite through a reaction with cadmium and reduced copper inside a reduction column. Reduced nitrite was then derivatized with Griess reagent, and the level of diazo compounds was analyzed by detection at 540 nm as previously described in detail [25].

### 2.7. N-Terminal Pro-Brain Natriuretic Peptide and Potassium in Plasma

Plasma levels of potassium and N-terminal pro-brain natriuretic peptide were analyzed according to an accredited hospital clinical laboratory procedure.

### 2.8. Cyclic Guanosine Monophosphate in Plasma

To prevent cyclic guanosine monophosphate (cGMP) degradation, an inhibitor of cAMP/cGMP phosphodiesterase; IBMX, 3-isobutyl-1-methylxanthine, 10 uM (I7018; Sigma-Aldrich, Merck, Sweden) was added to the plasma collected for cGMP measurement. Plasma cGMP concentration was analyzed using a commercially available ELISA (Cayman Chemicals #581021 BioNordika, Solna, Sweden), according to the manufacturer’s instructions.

### 2.9. Kinetic Analysis

The kinetic data were analyzed by non-compartmental methodology (NCA), using Phoenix^®^ WinNonlin^TM^ 6.4 from Certara (Princeton, NJ, USA). The maximal concentration, C_max_, and time at maximal concentration, t_max_, were direct observations from the concentration versus time data. The area under the plasma concentration versus time curve from time 0 to time-point at the last measurable concentration (AUC_last_) was calculated using the linear trapezoidal rule, where *C* is the concentration at the sampling time point *t*, and $\partial t=t_{2}-t_{1}$:$$AUC_{t_{1\;to\;}t_{2}}=\partial t\times \frac{C_{1}+C_{2}}{2}$$

The elimination rate constant, ʎ_z_, was determined by linear regression of the terminal portion of the concentration versus time profile. The terminal half-life, t_1/2_, was determined from ʎ_z_. The half-life was only calculated if at least 3 data points other than C_max_ could be included in the regression and R^2^ ≥ 0.80.
$$t_{1/2}=\frac{ln2}{{ʎ}_{z}}$$

AUC_last_ for nitrate and nitrite after ingestion of active BJ was corrected for AUC_last, placebo_ and for the body weight (BW) difference between the HD group and the HV:$$AUC_{last,\;corrected}=\left(AUC_{last,\;total}-AUC_{last,\;placebo}\right)\times \frac{BW,\;individual}{Mean\;BW,\;healthy\;volonteers}$$

Plasma C_max_ for nitrate and nitrite after ingestion of active BJ was corrected for the placebo concentration at the corresponding time-point and for the weight difference between the HD group and the HV:$$C_{max,\;corrected}=\left(C_{max,\;total}-C_{placebo}\right)\times \frac{BW,\;individual}{Mean\;BW,\;healthy\;volonteers}$$

### 2.10. Statistical Analyses

Clinical data of participant characteristics, HD treatment, as well as the effects of active BJ or placebo BJ, were analyzed using paired or unpaired parametric and non-parametric tests (i.e., *t*-test, Mann–Whitney test, and Wilcoxon matched-pairs signed rank test), as appropriate. Continuous data for nitrate, nitrite and cGMP levels were analyzed by using one-way or two-way analysis of variance (ANOVA) with repeated measures, followed by Tukey or Sidak post hoc test analysis of differences, as appropriate. Test for normal distribution (normality and lognormality tests) was performed by using the Shapiro–Wilk test or the D’Agostino and Pearson test. Phoenix^®^ WinNonlin^TM^ software (version Phoenix 6.4, Certara Inc., Princeton, NJ, USA) was used for the kinetic analysis of plasma nitrate and nitrite data. The geometric mean values and the 90% confidence intervals (CIs) for the ratio of the geometric mean values of C_max_ and AUC_last_ were also provided. Statistical analyses of pharmacokinetic data were performed using the Student’s *t*-test. Paired analyses were performed for comparisons between active and placebo within the same group, while unpaired analyses were performed for comparisons between HD patients and HV. All statistical tests were performed two-sided, and a *p*-value of less than 0.05 was considered statistically significant. A trend was noted at an alpha between 0.05 and 0.10. Statistical analyses were performed by using GraphPad Prism software (GraphPad Prism 9 for macOS, version 9.3.1, 2021, GraphPad Software Inc., San Diego, CA, USA).

## 3. Results

### 3.1. Characteristics of the Study Populations

The characteristics of the HD patients and the HV at the start of the study are summarized in Table 1. There were no differences in age or sex between the two groups. Body weight and height were significantly higher in HV as compared with HD patients, who displayed higher BP. All HD patients were receiving erythropoietin, B-complex vitamin, and folic acid.

Seven patients were on phosphate binders (sevelamer), and six patients were receiving calcium supplementation. The active form of vitamin D (calcitriol/paricalcitol) was prescribed in six patients and the calcimimetic cinacalcet was prescribed in four patients, respectively. Four patients were receiving proton pump inhibitors, four antihypertensive medication (beta blockers/calcium channel blockers), and two statins, and three patients received acetylsalicylic acid, respectively. Three patients were on calcineurin inhibitor and two were on corticosteroids.

Characteristics of HD patients before, at the end of dialysis session, and 4 h after nitrate-rich (active) and nitrate-depleted (placebo) BJ ingestion are summarized in Table 2. Blood pressure decreased at the end of dialysis. However, no significant changes in BP levels were observed 4 h after either active or placebo BJ ingestion. Characteristics of HV before and after active and placebo BJ ingestion are summarized in Table 3. Similarly to what was observed in HD patients, neither active nor placebo BJ intake were associated with significant changes in BP.

Overall, a single dose of active and placebo BJ was well tolerated both in HD patients and in HV during the study. An HV experienced an event of pain and swelling of the right arm after several attempts at venipuncture on the subject. Doppler ultrasound of the right upper extremity showed a partial thrombosis of basilic vein. The coagulation laboratory, including factor V Leiden mutation, prothrombin gene G20210A mutations, and antiphospholipid antibodies was negative. She was treated with anticoagulation therapy for three months. Follow-up Doppler ultrasound showed a complete resolution of the thrombosis. This serious adverse event was assessed as related to traumatic venipuncture.

### 3.2. Changes of NO Markers, Potassium, and Pro-BNP

In agreement with our previous studies in patients with HD [10], there was a significant increase in the concentrations of nitrate and nitrite in the spent dialysate (Appendix A), which confirm that the dialysis session is removing these NO metabolites from the body.

Plasma profiles for nitrate and nitrite in HD patients and in HV after the intake of active and placebo BJ are shown in Figure 1. In HD patients, the plasma levels of nitrate were significantly reduced after dialysis, whereas nitrite levels did not significantly change (Figure 1A,C). After the intake of active BJ, the nitrate and nitrite levels increased significantly. In HV, after the intake of active BJ, a similar, yet not as marked increase in plasma nitrate and nitrite levels was also observed (Figure 1B,D). In both groups, no significant changes were observed after the intake of placebo BJ (Figure 1A–D).

Similar to that observed for the plasma, the saliva levels trended to decrease during the dialysis albeit not significantly. After the intake of active BJ, but not placebo, the saliva nitrate and nitrite levels increased significantly in both HD patients and in HV (Appendix A).

Plasma levels of cGMP, potassium, and pro-BNP are shown in Figure 2. In HD patients, before the start of the dialysis session, the cGMP levels were significantly higher than in HV (Figure 2A,B). Dialysis significantly reduced the amount of cGMP to levels such as those observed in HV. Despite the elevated levels of circulating nitrate and nitrite observed after intake of active BJ, no significant increases in cGMP levels were noted.

At the end of dialysis session, cGMP levels tended to increase steadily regardless of the ingested BJ (Figure 2A). In HV, however, neither type of BJ showed significant changes in cGMP levels over time (Figure 2B).

As expected, plasma potassium levels decreased significantly at the end of dialysis (Figure 2C). Post dialysis, plasma potassium levels increased continuously regardless of the BJ ingested and reached similar levels as those observed before the start of the dialysis session (Figure 2C).

The release of Pro-BNP is influenced by changes in intracardiac pressure, and its circulating levels are commonly used to assess cardiac function [26,27]. Pro-BNP can also stimulate the release of cGMP via downstream signaling [26]. In our HD patients, pre-dialysis pro-BNP levels were well above normal reference values (Figure 2D). Pro-BNP levels at the end of dialysis were significantly lower than pre-dialysis levels (Figure 2D). After 44 h, pro-BNP levels were again similar to those observed at the start of dialysis (Figure 2D).

### 3.3. Kinetic Analysis

The kinetic analysis of plasma nitrate and nitrite is shown in Table 4 and Table 5, respectively. In HD patients, the intake of active BJ resulted in a 6.34-fold increase in nitrate AUC_last_ and a 6.61-fold increase in nitrate C_max_, whereas in HV, the increase in AUC_last_ and in C_max_ was 4.10- and 7.70-fold after active BJ ingestion, respectively (Table 4).

In HD patients, the terminal half-life of nitrate was approximately two-fold longer than in HV (33.5 vs. 15.2 h). Of note, taking into consideration the criteria for determination of the elimination rate constant, terminal half-life could not be estimated in four HD patients and in one HV. After the intake of placebo BJ, a significantly higher nitrate AUC_last_ was observed in HD patients as compared with HV. In HD patients, after the intake of active BJ, the nitrate t_max_ ranged from 45 min to 8 h with a geometric mean value of 2 h. In HV, the nitrate t_max_ ranged from 45 min to 4 h with a geometric mean value of 1.4 h (*data not shown*).

After the intake of active BJ as compared to placebo BJ, nitrite AUC_last_ increased 2.67-fold and 1.98-fold in HD patients and HV, respectively, whereas nitrite C_max_ increased approximately 3-fold in both groups (Table 5). In HD patients, after the intake of active BJ the nitrite t_max_ ranged from 45 min to 44 h, with a geometric mean value of 4.6 h. In HV the nitrite t_max_ ranged from 1.5 h to 8 h, with a geometric mean value of 3.7 h (*data not shown*).

Considering the difference in weight between HD patients and HV as well as the volume of BJ taken, the HD patients received a higher nitrate dose based on adjusted body weight than HV. Consequently, adjustments of the AUC_last_ and C_max_ values were performed, after subtraction of the placebo values, by using a body weight correction factor (see Materials and Methods). This adjustment resulted in slightly lower AUC_last_ and C_max_ differences between the groups after ingestion of the active BJ, although the corrected AUC_last_ for nitrate was still significantly higher in HD patients as compared with the HV group (Table 6). 

## 4. Discussion

This is the first study to examine the kinetics of plasma nitrate and nitrite concentrations after acute ingestion of nitrate-rich concentrated BJ in patients on maintenance HD. It is well established that NO plays key roles in the cardiovascular system and is also implicated in physiologic processes that influence both acute and long-term control of kidney function [3]. Indeed, abnormal levels of this signaling molecule have been associated with the development and progression of cardiorenal disorders [3]. In our body, NO derived enzymatically from the NOS system, which is rapidly oxidized to form the more stable anions nitrite and nitrate. Due to their longer half-life in circulation, these two anions have often been used to estimate the overall endogenous NO production. However, evidence accumulated over the last two decades has revealed that with the help of oral commensal bacteria and by various non-enzymatic and enzymatic processes (e.g., deoxyhemoglobin, deoxymyoglobin, xanthine oxidoreductase and mitochondrial complexes), nitrate and nitrite can undergo reductive conversion to form bioactive nitrogen species including NO [7]. This alternative route for NOS-independent NO generation, the nitrate–nitrite–NO pathway, can be boosted via our daily diet by consuming nitrate-rich food components such as green leafy salad and red beets [4,6].

Although the risk of having adverse cardiovascular events is already increased in patients with early stages of chronic kidney disease as compared with age-matched healthy subjects, patients with advanced chronic kidney disease as well as patients on dialysis exhibit a markedly elevated risk for cardiovascular morbidity and mortality [28]. Traditional cardiovascular risk factors such as hypertension, insulin resistance/diabetes, and dyslipidemia are highly prevalent in patients with chronic kidney disease. Importantly, all these risk factors have been associated with oxidative stress, reduced function of endothelial NOS and NO bioavailability [3,4]. In previous studies, we have observed that HD and peritoneal dialysis performed in adults and children lower circulating levels of nitrate and nitrite as well as markers of NO downstream signaling [9,10,29]. Given the important role of NO in maintaining cardiovascular homeostasis [4,6], these changes may contribute to an increased risk for CVD in this study population.

Recent dietary and pharmacological strategies that alleviate oxidative stress and restore NO bioavailability have been suggested to have therapeutic value in various reno-cardiovascular disorders including advanced chronic kidney disease and hypertension [3,4,8]. Long-term dietary supplementation with nitrate, aimed to stimulate the nitrate–nitrite–NO pathway, has been shown to have favorable effects on cardiovascular [6,21], metabolic [30], and renal functions [8] in experimental disease models, and to improve endothelial function and lower BP in hypercholesterolemic and hypertensive individuals [31,32]. Additionally, pilot studies showed that acute dietary nitrate load, using a similar concentrated BJ as in our study, lowered BP, reduced renal resistive index and improved exercise capacity in chronic kidney disease patients [33,34]. Whether this strategy could be used repeatedly to reduce the increased cardiovascular risk in individuals with mild to moderate chronic kidney disease and in HD patients, e.g., by lowering blood pressure, has not been studied.

When discussing the therapeutic value of dietary nitrate, it should be noted that the doses of inorganic nitrate often used in human dietary supplementation studies exceeds the Acceptable Daily Intake (ADI) for nitrate (i.e., 3.7 mg/kg BW/day). A concern has therefore been the potential formation of carcinogenic nitrosamines from nitrite in the acidic environment of the stomach [35]. Even though it seems evident that intake of preformed nitrosamines can be associated with some forms of cancer, dietary intake of inorganic nitrate, especially from natural sources, has not been shown to promote cancer [36,37]. This has unfortunately not yet led to a change in the ADI recommendations. It should also be mentioned that several types of diets associated with beneficial cardiovascular health effects, such as the Mediterranean and traditional Japanese diets, are high in inorganic nitrate, and adherence to these would in many cases exceed the ADI for nitrate. Taken together, we conclude that there is very little evidence to support that the repeated intake of a natural dietary source of nitrate should be considered carcinogenic.

Similar to that described in our previous studies in adult HD patients [10], the dialysis session was associated with the extraction of nitrate and nitrite from the body, as evident from the significantly increased levels of these anions in the spent dialysate. The amount of these NO metabolites that were lost during the dialysis equals the approximate amounts of NO being generated daily by endogenous NOS systems [38,39], which we [9,10,29] and others [40] have previously suggested contribute to the increased risk of cardiovascular complications associated with repeated dialysis.

In our study, we examined the kinetics of plasma nitrate and nitrite after a single intake of high inorganic nitrate in HD patients and show that this dietary approach with nitrate-rich concentrated BJ has been safe and well tolerated. Compared with HV, acute intake of nitrate was associated with significantly higher maximal plasma concentrations for nitrate and nitrite in HD patients as well as higher AUCs of these anions. Circulating nitrate (and nitrite) is mainly eliminated via the kidneys [41,42]. Therefore, the increased plasma concentrations of these two anions observed in our HD patients is most likely due to their full loss of residual renal function.

We have previously showed that cGMP levels are significantly reduced after dialysis sessions both in children and in adults [9,10]. In the current study, the cGMP levels were significantly higher in HD patients before the start of the dialysis session as compared with HV. As in our previous studies, the cGMP levels were reduced by dialysis. We noted, however, that cGMP levels did not change in parallel with the increase in plasma nitrate and nitrite concentration observed after active BJ ingestion. Instead, plasma levels of cGMP correlated with plasma changes in pro-BNP, which is normally released with increased ventricular filling pressure, and similar to NO, signals downstream via cGMP production [26]. It is noteworthy to mention that pro-BNP is dependent on kidney clearance for elimination, and therefore, kidney function has a well-documented inverse correlation with pro-BNP [43].

Because the kidney plays a critical role in controlling potassium balance, hyperkalemia is a common finding in patients with chronic kidney disease and has been associated with an increased risk for adverse cardiovascular events [44,45]. The association between plasma potassium levels and morbidity/mortality in HD patients has also been discussed [46,47]. The nitrate-rich BJ used in the current study contains relatively high levels of potassium (850 mg/70 mL), and therefore, we devoted particular attention to plasma concentration of potassium over time. In our study, as expected, plasma potassium levels decreased significantly after dialysis. We observed, however, that the increase in plasma potassium levels between dialysis sessions was similar regardless of the BJ ingested.

A well-documented effect following inorganic nitrate consumption observed by our research group is a decrease in BP [48], which was confirmed in subsequent studies [20,21,31]. This effect is, however, mostly observed after repeated nitrate consumption. Therefore, it is not unexpected that we did not observe any significant decrease in BP after a single intake of active BJ. Moreover, increased oxidative stress in the HD patients may have scavenged an expected increase in NO, and it may therefore be suggested that a higher dose of nitrate could be needed to achieve a BP-lowering effect in these patients.

## 5. Conclusions

In our kinetic analysis of dietary supplementation with nitrate, using concentrated BJ, we observe that acute single intake was safe and well tolerated, with significantly higher plasma concentrations of nitrate in HD patients as compared with HV. Long-term dietary nitrate supplementation efficacy studies are warranted to investigate the clinical benefits in subjects with reduced NO bioactivity, such as patients with chronic kidney disease, including HD patients.

## Figures and Tables

**Figure 1 nutrients-14-02480-f001:**
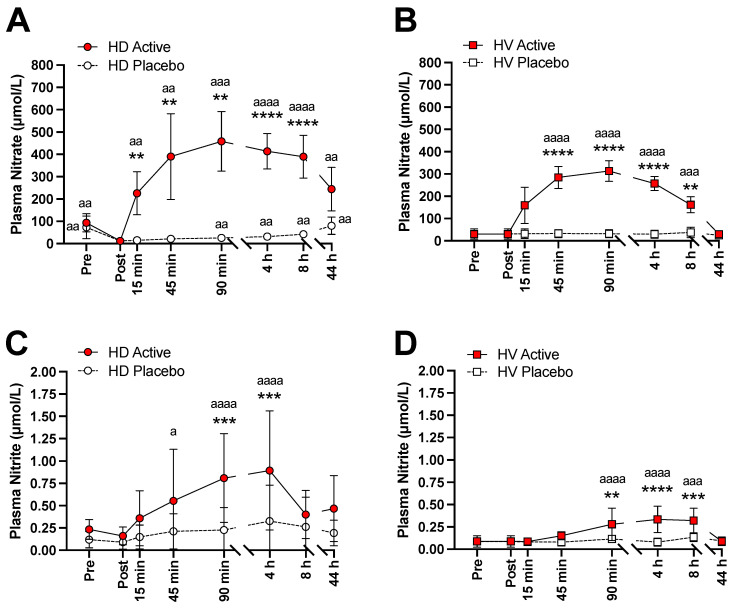
Plasma levels of nitrate and nitrite after single ingestion of nitrate-enriched (Active) and nitrate-depleted (Placebo) beetroot juice (BJ) in hemodialysis patients and in healthy volunteers. Legend: Plasma nitrate (**A**,**B**) and nitrite (**C**,**D**) levels in hemodialysis patients (HD, left panels) and in healthy volunteers (HV, right panels) after active BJ (filled red) or placebo BJ (open white) ingestion, respectively. HD, hemodialysis patients; HV, healthy volunteers. Values are presented as mean and standard deviation. ** *p* < 0.01; *** *p* < 0.001; **** *p* < 0.0001 as compared with placebo in the same group. ^a^
*p* < 0.05; ^aa^
*p* < 0.01; ^aaa^
*p* < 0.001; ^aaaa^
*p* < 0.0001 as compared with “post” values in HD patients or “baseline” values in HV, before active BJ and placebo BJ ingestion, respectively.

**Figure 2 nutrients-14-02480-f002:**
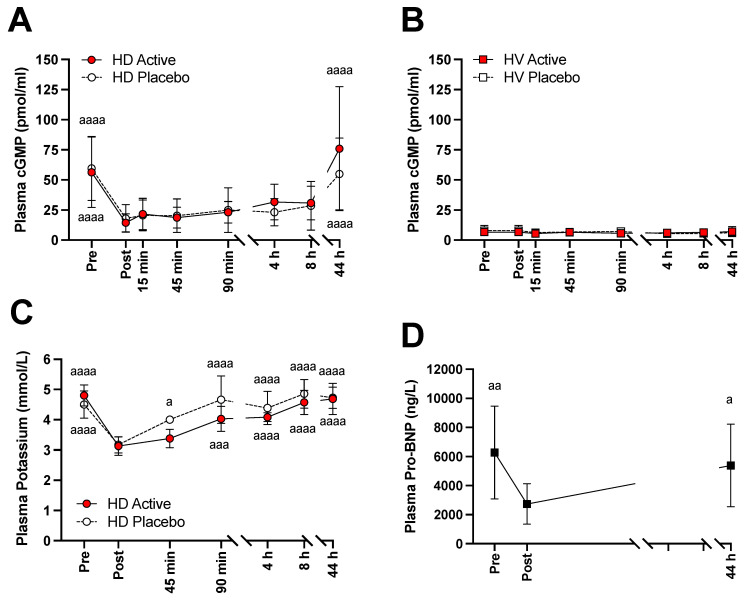
Plasma levels of cGMP, potassium, and pro-BNP after single ingestion of nitrate-enriched (Active) and nitrate-depleted (Placebo) beetroot juice (BJ) in hemodialysis patients and in healthy volunteers. Legend: Plasma cGMP levels in hemodialysis patients (HD, Panel (**A**)) and in healthy volunteers (HV, Panel (**B**)) after active BJ (filled red) or placebo BJ (open white) ingestion, respectively. Plasma potassium levels (**C**) in HD patients after active BJ (filled red) or placebo BJ (open white). Plasma pro-BNP levels (**D**) in HD patients without having received either active or placebo BJ. HD, hemodialysis patients; HV, healthy volunteers; pro-BNP, N-terminal pro-brain natriuretic peptide. Values are presented as mean and standard deviation. ^a^
*p* < 0.05; ^aa^
*p* < 0.01; ^aaa^
*p* < 0.001; ^aaaa^
*p* < 0.0001 as compared with “post” values in HD patients and “baseline” values in the HV, before active BJ and placebo BJ ingestion, respectively. Pro-BNP normal reference values: Healthy adults up to 74 yr: ≤125 pg/mL; >75 yr: ≤210 pg/mL. Healthy population 97.5 percentile: <44 yr: 125 pg/mL; 45–54 yr: 172 pg/mL; 55–74 yr: 350 pg/mL; >75 yr: 750 pg/mL.

**Table 1 nutrients-14-02480-t001:** Characteristics of participants at the start of the study.

**Characteristics**	**Hemodialysis Patients** (*n* = 8)	**Healthy Volunteers** (*n* = 7)	** *p* **
Age, years	31 (25.75–38.75)	36 (30–40)	0.380
Gender (male), n (%)	4 (50)	3 (42)	0.999
Body weight ^(a)^, kg	53.3 (50.0–62.6)	77.0 (60.3–84.2)	0.048
Height, m	1.60 (1.54–1.64)	1.67 (1.63–1.76)	0.025
Body mass index ^(b)^, kg/m^2^	20.9 (20.1–23.3)	25.4 (22.7–27.2)	0.302
SBP ^(c)^, mmHg	135 (119–147)	105 (101–110)	0.002
DBP ^(c)^, mmHg	85 (73–92)	65 (62–70)	0.020
Dialysis vintage, months	138.5 (75.75–201.5)	NA	−
Hypertension, n (%)	4 (50)	NA	−
Ischemic cardiomyopathy, n (%)	0 (0)	NA	−
Arrythmia, n (%)	0 (0)	NA	−
Peripheral vascular disease, n (%)	0 (0)	NA	−
Type 2 diabetes, n (%)	0 (0)	NA	−
COPD n (%)	1 (13)	NA	−
Previous renal Tx, n (%)	7 (88)	NA	−
Smoking, n (%)	2 (25)	NA	−

*Legend: COPD, chronic obstructive pulmonary disease; Tx, transplantation; NA, not applicable. ^(a)^ In participants on hemodialysis, it represents their first body weight after the dialysis session at the start of the study. ^(b)^ Body mass index was calculated using the body weight as described above. ^(c)^ All systolic (SBP) and diastolic (DBP) blood pressure values are the mean of the 2nd and 3rd reading. Values are presented as medians and interquartile ranges (IQR), unless otherwise stated. p values denote comparisons between hemodialysis patients and healthy volunteers*.

**Table 2 nutrients-14-02480-t002:** Characteristics of hemodialysis patients before the ingestion of nitrate-rich (Active) and nitrate-depleted (Placebo) beetroot juice (BJ).

Hemodialysis Patients ^(a)^			
Characteristics	Active BJ (*n* = 8)	Placebo BJ (*n* = 8)	*p*
Pre-dialysis body weight, kg	55.2 (52.7–64.5)	55.3 (52.9–65.4)	0.456
Post-dialysis body weight, kg	53.4 (49.4–62.6) ***	53.4 (50.6–62.6) ***	0.847
Pre-HD SBP ^(b)^, mmHg	137 (119–146)	128 (118–149)	0.675
Post-HD SBP, mmHg	114 (107–135) *	129 (106–138)	0.575
4 h post-HD SBP, mmHg	119 (106–128)	139 (116–140)	0.200
Pre-HD DBP ^(b)^, mmHg	85.5 (67.5–93.8)	80.5 (75.3–92.8)	0.929
Post-HD DBP, mmHg	64.5 (56.0–81.0) *	75.5 (58.0–85.0) *	0.729
4 h post-HD DBP, mmHg	75.0 (60.0–83.0)	81.0 (72.0–85.0)	0.212
HD session length, min	230 (209–239)	233 (209–249)	0.945
Blood flow rate, mL/min	383 (352–398)	391 (349–398)	0.813
Total volume ^(c)^, L	139 (136–142)	133 (122–146)	0.688
Ultrafiltration, mL	2332 (1734–3191)	2811 (1978–3233)	0.250
Kt/V_urea_	2.49 (1.65–2.79)	2.38 (1.73–2.80)	0.712
Kt/V_urea_ ≥ 1.2, n (%)	8 (100)	8 (100)	0.999
Residual urine, n (%)	0 (0)	0 (0)	0.999

*Legend: ^(a)^ All participants served as their own controls. SBP, systolic blood pressure; DBP, diastolic blood pressure; Kt/V_urea,_ index of adequacy of dialysis. ^(b)^ All SBP and DBP values are the mean of the 2nd and 3rd reading. ^(c)^ Total volume represents the total effluent volume at the end of the hemodialysis. The total effluent volume includes the total spent dialysate, ultrafiltrate, and the substitution volume during the treatment time. Values are presented as medians and interquartile ranges (IQR), unless otherwise stated. p values denote comparisons between active and placebo BJ in hemodialysis patients. * p < 0.05 and *** p < 0.001 between pre- and post-values for the same parameter within each intervention group*.

**Table 3 nutrients-14-02480-t003:** Characteristics of healthy volunteers before the ingestion of nitrate-enriched (Active) and nitrate-depleted (Placebo) beetroot juice (BJ).

Healthy Volunteers ^(a)^			
Characteristics	Active BJ (*n* = 7)	Placebo BJ (*n* = 7)	*p*
Body weight, kg	77 (60.30–84.20)	77 (60.30–84.20)	0.999
Baseline ^(b)^ SBP ^(c)^, mmHg	100 (100–110)	110 (102–110)	0.464
4 h post-baseline SBP, mmHg	100 (93–120)	105 (90–120)	0.803
Baseline ^(b)^ DBP ^(c)^, mmHg	70 (63–75)	64 (60–66)	0.123
4 h post-baseline DBP, mmHg	70 (60–70)	60 (60–70)	0.748

*Legend: ^(a)^ All participants served as own controls. ^(b)^ Baseline is regarded as the time-point before the ingestion of nitrate-enriched/placebo BJ. SBP, systolic blood pressure; DBP, diastolic blood pressure. ^(c)^ All SBP and DBP values are the mean of the 2nd and 3rd reading. Values are presented as medians and interquartile ranges (IQR), unless otherwise stated*.

**Table 4 nutrients-14-02480-t004:** Kinetic analysis of plasma nitrate after single ingestion of nitrate-enriched (Active) and nitrate-depleted (Placebo) beetroot juice (BJ) in hemodialysis patients and in healthy volunteers.

	**Active BJ**		**Placebo BJ**		**Active BJ/Placebo BJ**	
	**AUC_last_** **(h × µmol/L)**	**C_max_** **(µmol/L)**	**AUC_last_** **(h × µmol/L)**	**C_max_** **(µmol/L)**	**AUC_last_** **Ratio**	**C_max_** **Ratio**
Hemodialysis Patients *n =* 8	14300 *** (9150–20900)	485 *** (247–720)	2250 (1510–3540)	73.4 (45.5–133)	6.34 (5.52–7.28)	6.61 (4.75–9.19)
Healthy Volunteers *n =* 7	5640 ***/### (3810–7110)	316 ***/### (233–383)	1370 # (746–2470)	41.1 (19.7–81.8)	4.10 ## (3.21–5.24)	7.70 (5.26–11.3)

*Legend: Plasma nitrate AUC_last_, C_max_, in HD patients and in healthy volunteers after 70 mL concentrated BJ (active) or placebo BJ ingestion. Data are presented as geometric mean values. In brackets: Range for columns with Active and Placebo, and 90% Confidence Interval of AUC ratio of geometric mean for columns with Active/Placebo Ratio. Student’s paired t-test: *** p < 0.001 as compared with placebo. Student’s t-test: ^#^ p < 0.05; ^##^ p < 0.01; ^###^ p < 0.001 as compared with HD patients*.

**Table 5 nutrients-14-02480-t005:** Kinetic analysis of plasma nitrite after single ingestion of nitrate-enriched (Active) and nitrate-depleted (Placebo) beetroot juice (BJ) in hemodialysis patients and in healthy volunteers.

	Active BJ		Placebo BJ		Active BJ/Placebo BJ	
	AUC_last_ (h × µmol/L)	C_max_ (µmol/L)	AUC_last_ (h × µmol/L)	C_max_ (µmol/L)	AUC_last_ Ratio	C_max_ Ratio
Hemodialysis Patients *n =* 8	19.5 * (7.66–64.6)	0.862 * (0.346–2.13)	7.30 (2.96–31.1)	0.273 (0.106–1.25)	2.67 (1.82–3.94)	3.16 (2.00–4.99)
Healthy Volunteers *n =* 7	9.75 *** (4.40–14.2)	0.411 ***/# (0.216–0.619)	4.93 (2.39–7.67)	0.137 (0.0625–0.206)	1.98 (1.90–2.06)	3.01 (2.54–3.58)

*Legend: Plasma nitrite AUC_last_ and C_max_ in HD patients and in healthy volunteers after 70 mL concentrated BJ (active) or placebo BJ ingestion. Data are presented as geometric mean values. In brackets: Range for columns with Active and Placebo, and 90% Confidence Interval of AUC ratio of geometric mean for columns with Active/Placebo Ratio. Student’s paired t-test: * p < 0.05; *** p < 0.001 as compared with placebo. Student’s t-test: ^#^ p < 0.05 as compared with HD patients*.

**Table 6 nutrients-14-02480-t006:** Kinetic analysis of plasma nitrate and nitrite after single ingestion of nitrate-enriched (Active) and nitrate-depleted (Placebo) beetroot juice (BJ) in hemodialysis patients and in healthy volunteers.

	Plasma Nitrate		Plasma Nitrite	
Group	Corrected AUC_last_ (h × µmol/L)	Corrected C_max_ (µmol/L)	Corrected AUC_last_ (h × µmol/L)	Corrected C_max_ (µmol/L)
**Hemodialysis Patients** ***n =* 8**	9070 (6230–11800)	338 (185–548)	6.04 (0.414–25.5)	0.467 (0.233–1.24)
**Healthy Volunteers** ***n =* 7**	4090 ### (3020–6780)	282 (240–367)	4.73 (2.37–7.70)	0.301 (0.133–0.446)

*Legend: Plasma nitrate and nitrite AUC_last_ and C_max_ corrected for AUC_last, placebo_ and C_max, placebo_, respectively, and for body weight, in HD patients and in healthy volunteers after 70 mL concentrated BJ. Data are presented as geometric mean values and the range within the brackets. Student’s t-test: ^###^ p < 0.001 as compared with HD patients*.

## Data Availability

The data from the current study are not publicly available but are available from the corresponding authors upon request.

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
