# Peer review of "Plasma Nitrate and Nitrite Kinetics after Single Intake of Beetroot Juice in Adult Patients on Chronic Hemodialysis and in Healthy Volunteers: A Randomized, Single-Blind, Placebo-Controlled, Crossover Study"

_nutrients, 2022, doi:10.3390/nu14122480_

Round 1
Reviewer 1 Report
The authors have adequately described a single-dose experiment. The authors have adequately connected the process of hemodialysis to the reduction of measured outcome variables, which would be expected with standard treatment. The authors have not over estimated the significance of this observation, Therefore, the report is explained well although only an exploratory report with no proof (shown or claimed) of clinical application.
Author Response
We thank the reviewer for the positive comments. The manuscript has been revised taking into consideration the reviewers’ comments and suggestions.
Reviewer 2 Report
The manuscript shows the results of a randomized, participant-blind, cross-over design clinical trial. Two groups, one with patients with chronic kidney disease undergoing hemodialysis and a group of healthy participants were randomized to take a beet-root extract rich in nitrates (400 mg) or placebo (extracted with 0 mg of nitrates). The study aimed to monitor the kinetics of nitrites and nitrates following ingestion.
As expected, immediately after dialysis, patients showed a drop in nitrate concentrations and a sudden increase in case of the intervention, compared to placebo. This increase was more evident in hemodialysis patients than in healthy controls, even in the case of weight correction (being the dialysis group with higher BMI). No effect on blood pressure was detected.
The manuscript is well written, the description of the experimental design is very detailed, and the supporting literature is up to date. The introduction is well focused on the topic, which concerns the possible negative effects of nitrite and nitrate depletion on cardiovascular and renal health following hemodialysis sessions.
I have some considerations:
- The number of participants is small. How was assessed for the enrollment phase?
- The study is based on the kinetics of nitrites and nitrates after beet-root juice ingestion, although an evaluation of secondary endpoints would have been useful to link the hypothesis to a possible benefit. For example, even if no changes in blood pressure were observed, although disappointing, is easily explained by the limits of the intervention (single ingestion, reduced times, reduced dosages). Do the authors think that the recruitment of patients with pre-existing changes in blood pressure may have influenced this result?
- Why the study was participant-blind and not double-blind?
- The possible benefits of providing nitrites and nitrates are well explained, but can't there also be negative aspects? For example, there is much discussion about the effect of these compounds on the formation of nitrosamines and methemoglobin. That has led to the definition of an ADI for nitrates of 3.7 mg/kg BW/day to limit the use of these compounds in processed foods. This amount is lower than that used in the trial.
Minor aspects:
- In figure 1C, “HDF”?
- In table 4, the asterisk in 1370 should perhaps be a hash?
Author Response
AUTHORS’ GENERAL RESPONSE: We would like to thank the reviewer for the careful and insightful discussion of our manuscript. Please, find below our responses to each of the comments. The revised version of the manuscript has been reworked accordingly.
I have some considerations:
- The number of participants is small. How was assessed for the enrollment phase?
AUTHORS’ RESPONSE: Thank you very much for your observation. Indeed, the reviewer is correct. However, this was a study aimed specifically at obtaining reliable information on nitrite and nitrate kinetics after single ingestion of beet-root juice (to probably be used in subsequent prospective controlled dietary intervention studies). As mentioned in our initial submission, we truly believe that this information is important to permit the design of well-controlled, scientifically valid intervention studies intended to investigate specific questions, such as a clinical outcome (e.g., decrease in blood pressure) or a laboratory outcome measured in an individual after randomization. This type of clinical study would necessarily require a much larger study population.
We would like to emphasize that we focused our investigation on nitrate and nitrite kinetic after only a single ingestion of beetroot Juice/placebo. Against this background, and considering that phase I single dose PK clinical research studying investigational new drugs is usually conducted in a restricted number of healthy volunteers/patients (European Medicines Agency, Pharmacokinetic Studies in Man, 3CC3a), that our kinetic data are crucial for planning further larger studies, and that our study has not been designed to assess efficacy and safety of beetroot Juice, we believe that the number of subjects included in this investigation should be considered acceptable for the purpose of the study.
- The study is based on the kinetics of nitrites and nitrates after beet-root juice ingestion, although an evaluation of secondary endpoints would have been useful to link the hypothesis to a possible benefit. For example, even if no changes in blood pressure were observed, although disappointing, is easily explained by the limits of the intervention (single ingestion, reduced times, reduced dosages). Do the authors think that the recruitment of patients with pre-existing changes in blood pressure may have influenced this result?
AUTHORS’ RESPONSE: Thank you very much for this observation, which is indeed a relevant one. As mentioned above, this is a kinetic study conducted in a small study population. Hence, we cannot draw any conclusions by linking the hypothesis to any secondary outcomes.
It can be hypothesized that favorable cardiovascular effects might be observed in a larger study population after repeated dietary nitrate supplementation. It can also be suspected that this type of dietary approach with inorganic nitrate would be more effective in patients with less severe CKD (Stage 2-4). However, at this early stage of our investigation, we do not have a clear picture about this, and therefore we would like to avoid speculations. In the section “Discussion” we have added the following phrase:
“Whether this strategy could be used repeatedly to reduce the increased cardiovascular risk in individuals with mild to moderate CKD and in HD patients, e.g., by lowering blood pressure, has not been studied.”
We hope the reviewer finds this wording appropriate.
- Why the study was participant-blind and not double-blind?
AUTHORS’ RESPONSE: We fully agree with this reviewer that blinding is a fundamental tool in clinical trial design and a powerful method for preventing and reducing bias. In the current study, the participant-blind design was chosen based on the following facts: i) even though the nitrate-rich and the nitrate-depleted beetroot juices that the participants received after randomization were identical in taste and appearance, the bottles have a mark (A or B), which allowed the investigator only to disclose the type of juice content; ii) we did not investigate specific clinical and/or a laboratory outcome(s) measured after repeated intake of beetroot juice/placebo in an individual after randomization. Therefore, the authors believe that the interpretation of our results is not hampered by the study design.
We are however aware that in prospective interventional studies, which are specifically tailored to evaluate the direct impact of a treatment, if the trial investigator is blinded, it is less likely that the investigator transfers inclinations to study participants, adjusts a dose, withdraws study participants, or encourages participants to continue participation.
- The possible benefits of providing nitrites and nitrates are well explained, but can't there also be negative aspects? For example, there is much discussion about the effect of these compounds on the formation of nitrosamines and methemoglobin. That has led to the definition of an ADI for nitrates of 3.7 mg/kg BW/day to limit the use of these compounds in processed foods. This amount is lower than that used in the trial.
AUTHORS’ RESPONSE: The reviewer brings up a concern that has been shared by other researchers (Zamani, H. et al., Crit Rev Food Sci Nutr 2021), when discussing the therapeutic value of dietary nitrate and we have added a paragraph about this in the revised discussion of the manuscript. The formation of carcinogenic nitrosamines from processed food or from nitrite in the acidic stomach has been extensively studied. Even though it seems evident that intake of preformed nitrosamines can be associated with some forms of cancer, dietary intake of inorganic nitrate, especially from natural sources, has not been shown to promote cancer (EFSA 2008, WHO 2003). This has unfortunately not led to a change in the ADI recommendations mentioned by the reviewer. It should also be mentioned that several diets shown to be associated with beneficial health effects, such as the Mediterranean and the traditional Japanese diets, which are high in inorganic nitrate. Moreover, adherence to these types of diets would in many cases exceed the ADI for nitrate.
Regarding methemoglobinemia, this will not occur from high nitrate intake in adults. However, it can occur if a nitrate-containing product is contaminated with bacteria that can efficiently reduce nitrate to nitrite.
Taken together, there is lack of evidence to firmly support that a natural dietary source of nitrate should be considered carcinogenic or increase the risk of methemoglobinemia. In the revised discussion of the manuscript, we have included the following paragraph about this particular issue:
“When discussing the therapeutic value of dietary nitrate, it should be noted that the doses of inorganic nitrate often used in human dietary supplementation studies exceeds the Acceptable Daily Intake (ADI) for nitrate (i.e., 3.7 mg/kg BW/day). A concern has therefore been the potential formation of carcinogenic nitrosamines from nitrite in the acidic environment of the stomach [35]. Even though it seems evident that intake of preformed nitrosamines can be associated with some forms of cancer, dietary intake of inorganic nitrate, especially from natural sources, has not been shown to promote cancer [36,37]. This has unfortunately not yet led to a change in the ADI recommendations. It should also be mentioned that several types of diets associated with beneficial cardiovascular health effects, such as the Mediterranean and traditional Japanese diets, are high in inorganic nitrate and adherence to these would in many cases exceed the ADI for nitrate. Taken together, we conclude that there is very little evidence to support that repeated intake of a natural dietary source of nitrate should be considered carcinogenic.”
References:
- Zamani, H. et al. The benefits and risks of beetroot juice consumption: a systematic review. Crit Rev Food Sci Nutr 2021, 61, 788-804.
- EFSA (2008). Nitrate in vegetables: Scientific Opinion of the Panel on Contaminants in the Food Chain. EFSA J. 689, 1–79.
- Speijers, G.J.A., and van den Barandt, P.A. (2003). Nitrate (and Potential Endogenous Formation of N-nitroso Compounds). WHO Food Additives Series 50. https://inchem.org/documents/jecfa/jecmono/v50je06.htm
Minor aspects:
- In figure 1C, “HDF”?
AUTHORS’ RESPONSE: Thank s very much for noticing this mistake, which has been corrected in the revised version of the manuscript (Figure 1).
- In table 4, the asterisk in 1370 should perhaps be a hash?
AUTHORS’ RESPONSE: The reviewer is correct, thank you! This has been corrected in the revised version of Table 4.
Round 2
Reviewer 2 Report
Thank you for your revisions. No further comment.